# Anti-Inflammatory Action of Resveratrol in the Central Nervous System in Relation to Glucose Concentration—An In Vitro Study on a Blood–Brain Barrier Model

**DOI:** 10.3390/ijms25063110

**Published:** 2024-03-07

**Authors:** Justyna Komorowska, Mateusz Wątroba, Małgorzata Bednarzak, Anna D. Grabowska, Dariusz Szukiewicz

**Affiliations:** Laboratory of the Blood-Brain Barrier, Department of Biophysics, Physiology and Pathophysiology, Medical University of Warsaw, Chalubinskiego 5, 02-004 Warsaw, Poland; lek.justyna.komorowska@gmail.com (J.K.); mateusz.watroba@wum.edu.pl (M.W.); anna.sepulveda@wum.edu.pl (A.D.G.)

**Keywords:** blood–brain barrier, astrocytes, glucose, resveratrol, inflammatory response modulation, anti-inflammatory action

## Abstract

Unbalanced blood glucose levels may cause inflammation within the central nervous system (CNS). This effect can be reversed by the action of a natural neuroprotective compound, resveratrol (RSV). The study aimed to investigate the anti-inflammatory effect of RSV on astrocyte cytokine profiles within an in vitro model of the blood–brain barrier (BBB) under varying glucose concentrations (2.2, 5.0, and 25.0 mmol/L), corresponding to hypo-, normo-, and hyperglycemia. The model included co-cultures of astrocytes (brain compartment, BC) and endothelial cells (microvascular compartment, MC), separated by 0.4 µm wide pores. Subsequent exposure to 0.2 μM LPS in the brain compartment (BC) and 50 μM RSV in the microvascular compartment (MC) of each well was carried out. Cytokine levels (IL-1 α, IL-1 β, IL-2, IL-4, IL-6, IL-8) in the BC were assessed using a Multi-Analyte ELISArray Kit before and after the addition of LPS and RSV. Statistical analysis was performed to determine significance levels. The results demonstrated that RSV reduced the concentration of all studied cytokines in the BC, regardless of glucose levels, with the most substantial decrease observed under normoglycemic conditions. Additionally, the concentration of RSV in the BC was highest under normoglycemic conditions compared to hypo- and hyperglycemia. These findings confirm that administration of RSV in the MC exerts anti-inflammatory effects within the BC, particularly under normoglycemia-simulating conditions. Further in vivo studies, including animal and human research, are warranted to elucidate the bioavailability of RSV within the central nervous system (CNS).

## 1. Introduction

Impaired glucose tolerance (IGT) and diabetes occur worldwide on a pandemic scale, which is promoted by increased metabolic disorders in addition to a rise in obesity levels. Because nervous cells preferentially use glucose to obtain energy, and the influx of glucose to neurons is not insulin dependent [1], oscillations of glucose concentration can affect metabolic processes within the CNS. Both diabetes-related hyperglycemia and iatrogenic hypoglycemia (induced by anti-diabetic insulin treatment) may affect the functioning of the CNS [2,3]. Inflammation in the CNS can damage the neurons, as was documented with Alzheimer’s disease, Parkinson’s disease, encephalitis, meningitis, neurosarcoidosis, and hyperglycemia [3,4,5,6,7,8,9]. Neuroinflammatory processes can be particularly difficult to predict in diabetes because the cytokine profile in the CNS depends on serum glucose concentration [10]. High prevalence of neuroinflammation-associated diseases impairs the functioning of millions of people all over the world. Therefore, many research studies aim at finding a remedy for neuroinflammation, including diabetes-associated neuroinflammation.

Hyperglycemia can result in an increased glucose influx to BBB-forming cells (both pericytes and astrocytes) [3,11]. Temporary, incidental high blood sugar levels are not harmful to neurons, because the passage of glucose into the cerebrospinal fluid is tightly regulated by a healthy BBB, but prolonged periods of hyperglycemia in diabetic patients can lead to the production of reactive oxygen species (ROS) in astrocytes and pericytes, primarily due to an accelerated process of aerobic oxidative glycolysis, followed by subsequent reactions in the tricarboxylic acid cycle that convert oxidized nicotinamide-adenine dinucleotide (NAD+) into its reduced form, NADH [12,13,14]. It should be noted that neither NAD+ nor NADH can cross the mitochondrial membrane, so mitochondrial NAD+ depletion cannot be neutralized by increased influx of NAD+ from other cell compartments and excess NADH cannot be eliminated from the mitochondrion through simple diffusion [15,16]. Electrons carried by NADH are transferred to the electron transport chain (ETC) to produce ATP, but this process can be ineffective when the ATP level in the cell is already high. In such circumstances, NADH can transfer electrons non-enzymatically to molecular oxygen, which results in ROS production [17]. Elevated levels of ROS in astrocytes and pericytes may trigger the activation of nuclear factor-κappa B (NF-κB), leading to inflammation and an increase in vascular permeability [18,19,20]. At this stage, glucose influx into the CNS ceases to be strictly controlled, which results in increased glucose influx into neurons [20,21], and induces an inflammatory response in neurons through activation of the NF-κB pathway described above. In addition, mtDNA damage by increased amounts of ROS can induce production of chaperon proteins, such as HSP-60 [22]. Some HSP-60 molecules can be transferred to other cell compartments, such as cell membranes, where they can activate pattern recognizing receptors (PRRs), which can, in turn, activate innate immunity-related inflammation [23]. Additionally, innate immunity can be also activated by advanced glycation products (AGEs) [24]. 

Since the brain does not produce glucose or store substantial glycogen reserves within astrocytes, it depends on a continuous influx of glucose from the bloodstream [25]. Consequently, hypoglycemia instigates the depletion of cerebral energy reservoirs, leading to cognitive malfunction of the brain, culminating in coma or death [26]. There is a plausible link between recurring instances of profound hypoglycemia and enduring cognitive impairment [27]. While the precise mechanism remains elusive, it has been documented that hypoglycemia leads to elevated levels of inflammatory cytokines and an increase in white blood cell count, indicating a potential association between low blood sugar and inflammation [28,29]. In the recently published study on induction of hypoglycemia in type 2 diabetes (T2D), the levels of both inflammation and oxidative stress markers were significantly increased in urine and blood samples at 24 h following the hypoglycemic episode in humans. This effect was not observed in T2D-free controls [30]. The above-described proinflammatory effects of metabolic disturbances in the CNS accompanying hyperglycemia and hypoglycemia are presented in Figure 1.

Resveratrol, a polyphenol naturally found in the skin of red grapes and berries, is known for its role in mitigating chronic inflammatory conditions [33]. Beneficial effects of RSV have been recently documented, e.g., in a diabetic elderly female rat model, in which RSV treatment resulted in cardioprotective activity [34,35]. The anti-inflammatory and antioxidant characteristics of RSV are associated with its ability to activate sirtuins (SIRTs) either directly or indirectly [36]. SIRTs activation occurs through allosteric modification of their molecules or by increase in the expression and activity of nicotinamide phosphoribosyl transferase (NAmPRTase or NAMPT) and AMPK [37,38]. SIRTs represent a family of enzymes, primarily functioning as NAD^+^-dependent deacetylases, consisting of seven members in humans and other mammals [39,40]. Activation of SIRTs may be responsible for neuroprotective effects of RSV that correspond to epigenetic inhibition of the p53 protein and epigenetic increase in the expression of peroxisome proliferator-activated receptor-gamma coactivator 1α (PGC-1α) [38,41,42]. Overall, the bioactivity of RSV as the neuroprotective agent includes inhibition of the NF-κB signaling pathway, activation of autophagy instead of apoptosis, and modulation of the key activity of mitogen-activated protein kinases (P38-MAPK), extracellular signal-regulated kinase 1/2 (ERK1/2), and phosphoinositide 3-kinase (PI3K)/Akt. [38,43]. Regardless of the not yet fully understood mechanisms of action, it has been confirmed in many research studies, performed mainly on in vivo models, that RSV shows beneficial effects on cognitive processes and memory, as well as on hippocampus connectivity and microstructure in older adults [44,45,46,47].

The BBB comprises a unique composition of circulatory system and CNS. It protects neurons from potentially harmful substances through passive and selective transport of ions, molecules, and cells from the blood to the cerebrospinal fluid (CSF) [48]. Endothelial cells within the BBB are strictly connected with occlusion zones, without fenestrations [49]. The maximum pore size within the BBB that allows for the passive flow of molecules is typically below 1 nm in diameter [50]. Therefore, the larger substance transport from the plasma occurs transcellularly through facilitated diffusion or the protein transport system (PTS) [51]. 

Thus, not all substances can cross the BBB and enter the CNS. Only molecules of molecular weight below 500 Da can effectively penetrate the BBB [52,53,54]. Numerous anti-inflammatory compounds are under investigation for their ability to cross the BBB and to reach the CNS. RSV, due to its low molecular weight of 228 Da and its lipid-soluble properties, can effectively cross the BBB [55].

Before the substances become available to neurons, they are transported into astrocytes [56]. In the human brain, astrocytes are much more numerous than neurons, and they have essential functions within the CNS, including regulation of the level of substances like glutamate, ions (such as calcium and potassium), and water [57]. Astrocytes also protected against damage from oxidative and nitrosative stress, store energy, assist in generating mitochondria, contribute to scar formation, aid in tissue repair through processes like angiogenesis and neurogenesis, and modulate synapses [58]. Additionally, as part of the BBB, astrocytes are strategically positioned between the blood vessels in the brain and the connections between neurons, allowing them to facilitate the uptake of nutrients from the bloodstream [59].

In the cytoplasm of the endothelial cells and astrocytes, substances are enzymatically processed to constitute another biochemical component of the BBB [60]. Interestingly, direct cell-to-cell contact between astroglial cells and brain capillary endothelial cells is the necessary precondition for normal enzymatic activities of gamma-glutamyltranspeptidase (γ-GTP) and alkaline phosphatase (ALP), the enzymes commonly used as markers to evaluate the BBB’s characteristics [61].

Due to the dependence of neuroinflammation on the body’s metabolic state, it is important to investigate whether the metabolic status, particularly in terms of plasma glucose levels, can alter the anti-neuroinflammatory effects of RSV. This study is a continuation of the analysis of changes in the cytokine profile under the influence of RSV at different glucose concentrations, based on an in vitro BBB model, the results of which have already been published in relation to interleukins (IL-10, IL-12, IL-17A), tumor necrosis factor α (TNF-α), interferon gamma (IFN-γ), and granulocyte-macrophage colony-stimulating factor (GM-CSF) [62]. The current analysis considers other cytokines (interleukins) with model neuroinflammatory effects (IL-1α, IL-8) [63,64], pro-inflammatory and neurodifferentiating potential (IL-1β) [65,66], neuro-immune-modulatory properties (IL-2) [67,68], and anti-inflammatory or neurotrophin-like activities within the CNS (IL-4, IL-6) [69,70,71]. 

## 2. Results

### 2.1. Co-Cultured Cells Viability and Morphology

Throughout the entire 36 h incubation period following the creation of the BBB model for the three study groups under hypo-, normo-, and hyperglycemic conditions, the consecutive intervals of 36 h after administering LPS had no significant impact on the vitality of co-cultured cells. This assessment was based on the results of the exclusion test using trypan blue and the observation of co-cultured cell morphology. Astrocytes after glucose and LPS administration are presented in Figure 2.

### 2.2. Effect of Glucose Concentration in MC on Inflammatory Response within BC

The mean concentration of measured cytokines in the samples obtained within the three studied groups (hypo-, normo-, and hyperglycemic) from the BC after 24 h following glucose addition, which may correspond with CSF in the BBB model used, are presented in Figure 3. 

The concentration of IL-1α in the samples from a high-glucose environment (group III) was the highest, measuring 0.915 ± 0.09 (pg/mL ± SD), while slightly lower values (0.89 ± 0.09) were observed in samples collected from a low-glucose environment (group I). Notably, the normoglycemic group (group II) exhibited the lowest mean IL-1α level, with a value of 0.58 ± 0.06, which was significantly different (*p* < 0.05) from the mean values in both group I and group III. 

The levels of IL-1β and IL-2 were not affected by different glucose concentrations, and the differences for these cytokines in hypo-, normo-, and hyperglycemia were not statistically significant. 

Interestingly, and unlike for the other studied cytokines, in the cytokine profile of IL-4, the highest level of IL-4 was observed in conditions imitating hypoglycemia, whereas the lowest IL-4 concentration was measured in the samples obtained from the hyperglycemic environment. These values significantly differed (*p* < 0.05) from the IL-4 level obtained for normoglycemia (see Figure 3). 

Leaving the cytokine concentrations, the profiles for IL-6 and IL-8 changed similarly, with statistically significant (*p* < 0.05) differences between the groups. Here, the lowest mean values were observed in the normoglycemic environment, intermediate levels in hypoglycemia, and the highest levels in hyperglycemia. The biggest difference measured in samples containing different glucose concentrations was shown within the IL-6 values, which amounted to 3.67 ± 0.37 (pg/mL ± SD) for normoglycemia and were increased to 7.68 ± 0.77 in the hyperglycemia-mimicking environment.

There was a big difference measured in the samples containing different glucose concentrations for the IL-8 values, which equaled about 7.2 ± 0.72 (pg/mL ± SD) when the glucose concentration was normal, while they rose to 9.84 ± 0.99 when the glucose concentration was high. This is in accordance with the existing results of other studies showing that hyperglycemic conditions increase the concentration of IL-8 in systems other than the BBB, such as in human gingival epithelial cells [72] or in keratinocytes [73], among others. It is worth noting that the results in the control group without glucose were undetectable.

### 2.3. Effect of LPS Concentration on Inflammatory Response within BC

Administration of 0.2 μM lipopolysaccharide (LPS) solution into the BC caused a wide variety of changes in the cytokine profiles within 12 h compared to the controls (Figure 4).

The levels of IL-1α, IL-1β, and IL-4 in groups I–III were similar to those in the respective control samples, indicating a lack of LPS-induced inflammatory response, whereas this response was observed for IL-6 and IL-8 with significant (*p* < 0.05) increases in these cytokine levels in all three examined groups. Moreover, for IL-6, the differences between both hypoglycemic + LPS and hyperglycemic + LPS subgroups versus normoglycemic + LPS subgroup were also significant. 

In the case of IL-2, the levels of the cytokine were indeed significantly increased after 12 h following LPS administration in group I and II (hypoglycemic and normoglycemic environment, respectively), but in hyperglycemic conditions (group III), the differences did not reach statistical significance.

### 2.4. Effect of RSV Administration to MC on Inflammatory Response within BC

After 36 h following LPS administration, the concentrations of the studied pro-inflammatory cytokines increased significantly (*p* < 0.05) in the majority of cases compared to the respective concentrations after 12 h (Figure 5). This was true for hypo-, normo-, and hyperglycemia, with a few exceptions: there was a lack of statistical significance between observed differences for IL-1α, IL-2 and IL-4 in the hypoglycemic environment and non-significant differences for IL-2 in hyperglycemia.

The effects of RSV administration on the cytokine profiles at different glucose concentrations are presented in Figure 6. 

In this stage of the experiment, separate tests for each glucose environment to compare the levels of six different cytokines 24 h after applying RSV to the cells in two groups: one with RSV and one without, were conducted. The Mann–Whitney U test for this comparison was used.

The null hypothesis for this test stated that the concentration of each specific cytokine would be identical 24 h after applying RSV in both the RSV group and the group without RSV, while the alternative hypothesis suggested that the presence of RSV would lead to a decrease in the concentration of these cytokines.

Furthermore, the percentage reduction in cytokine concentration levels at 24 h after administering RSV in three different study groups was assessed. The percentage drop (representing the effect of RSV) was determined by comparing each group to the one without the RSV solution.

The Mann–Whitney U test was used for this comparison as well. The null hypothesis for this test posited that the concentration of each specific cytokine 24 h after applying RSV would be the same regardless of the glucose level, with the alternative hypothesis suggesting that the normal glucose level would result in a greater reduction in cytokine levels.

The introduction of RSV into the MC led to a decrease in the levels of all cytokines in the BC after 24 h, irrespective of glucose levels, when compared to the groups that did not receive RSV. However, it is worth noting that not all of these changes reached statistical significance. 

RSV decreased the concentration of IL-1α by approximately 10%, regardless of glucose concentration, but the results were not statistically significant (*p* > 0.05). For IL-1β, this action of RSV was statistically significant; however, it was much stronger at normal glucose concentrations, where a major reduction in this cytokine was observed (the decrease in the IL-1β level was estimated to be 10 times), compared to hypoglycemia- or hyperglycemia-mimicking conditions, where the IL-1β concentration was 2 to 3 times lower after RSV administration. 

RSV significantly decreased IL-2 concentration only in normoglycemia-simulating conditions, by 29%. The results for IL-4 suggest that the optimal action of RSV towards the decreasing IL-4 level took place in hypoglycemic- and hyperglycemic conditions but not in normoglycemia. 

The levels of IL-6 and IL-8 decreased significantly 24 h after RSV administration into the MC. For IL-6, RSV reduced its concentration in hypo-, normo-, and hyperglycemic conditions by 42%, 53%, and 49%, respectively, and for IL-8 by 25%, 43%, and 31%. These results are in agreement with other studies that documented the ability of RSV to decrease both IL-6 and IL-8 levels [74,75,76].

Thus, the preliminary assumption that RSV added to MC reduces the concentration of cytokines in the BC after 24 h of application in the experimental environment regardless of glucose level was confirmed with statistical significance for measurements marked with an asterisk (*) in Figure 6. The largest decrease for IL-1β, IL-6, and IL-8 was observed in normoglycemia-simulating conditions compared to both hypo- and hyperglycemia (*p* < 0.05).

Referring to the numerical data in Figure 6, RSV administration in hypoglycemia-simulating conditions resulted in higher concentrations of IL-1β (+48.3%), IL-2 (+3.6%), IL-6 (+40%), and IL-8 (+32.2%), compared to the normoglycemic conditions, whereas the levels of other cytokines studied, i.e., IL-1α and IL-4, were lower in hypoglycemia- than in normoglycemia-mimicking conditions (−38.5% and −27%, respectively).

RSV administration in hyperglycemia-simulating conditions resulted in higher concentrations of IL-1β (+48.5%), IL-2 (+12.3%), IL-4 (+19.4%), IL-6 (+46.3%), and IL-8 (34.7%), compared to the normoglycemic conditions, while the level of IL-1α was lower in the hyperglycemic than in normoglycemic conditions (−4.7%).

### 2.5. RSV Penetration through the BBB Model Applied

An examination of RSV transfer between the two compartments through the semi-permeable membrane of the BBB model reveals that RSV exhibits limited bioavailability. Specifically, when 2282.5 ng of RSV was administered to the MC in a 200 µL (0.2 mL) volume, the resulting concentration of RSV in the MC was 11,412.5 ng/mL. Employing a well-established formula for calculating molar concentration and adhering to the instructions provided with the ELISA kits, the concentration of RSV in the BC was determined. The findings from these ELISA measurements and subsequent computations are presented in Figure 7. RSV concentrations in the BC are significantly lower than the initial amount administered in the MC for each glycemic condition. 

In this section of the experiment, the Mann–Whitney U test for paired samples was employed to assess the null hypothesis, which posited that there were no distinctions in the mean values of RSV concentration between populations exhibiting normoglycemia and hypoglycemia, as well as normoglycemia and hyperglycemia. Conversely, the alternative hypothesis posited that RSV would exhibit different levels in the BC in hypoglycemic and hyperglycemic conditions, in regard to normoglycemia. Each determination was conducted six times, and the outcomes are reported as mean values accompanied by their corresponding standard deviations.

After conducting six measurements, it was observed that the concentration of RSV in the BC was higher in normoglycemia- (0.61% of the initial concentration) compared to hypoglycemia- and hyperglycemia-mimicking conditions (0.53% and 0.55% of the initial concentration, respectively).

In the normoglycemic condition, even after 24 h of administering RSV to the MC, its concentration in the BC remained consistently high, significantly higher than in other groups (*p*-value < 0.02). In the hypoglycemic and hyperglycemic environment, the concentration of RSV measured 24 h after administration was significantly lower compared to the measurement taken after 12 h, dropping by more than 17 ng/mL and by 15 ng/mL, respectively.

These results confirm the initial hypothesis that abnormal blood glucose levels can significantly impair RSV availability in the CNS.

## 3. Discussion

Numerous variants of the BBB models have been created by different teams of researchers, with some being commercially available [77]. Diverse BBB models encompass various cell combinations, such as co-cultured pericytes, astrocytes, and endothelial cells, astrocytes with endothelial cells, or solely endothelial cells. Furthermore, in the realm of predicting BBB permeability, models can be categorized into monoculture, non-contact co-culture, and contact co-culture systems. Another classifying factor for BBB models is the species of origin of the cells applied, which includes animal models (e.g., mouse, rat, bovine, or porcine) and human models [78,79]. 

In our study, the BBB model was set up from human astrocyte and endothelial cell lines, to imitate conditions occurring on both sides of the barrier—namely, BC and MC. The advantage of such a model is the selectivity regarding the permeation test that is not influenced by cellular and non-cellular blood components or CSF [80]. On the other hand, such an approach and the use of cell lines has certain limitations, especially with regard to in vivo conditions.

The obtained results indicate that the abnormal level of glucose in the MC are already a trigger of an inflammatory reaction in the BC, reflected by the increased level of pro-inflammatory cytokines. It is important to note that both low and high blood sugar environments can cause inflammation, as has been observed in previous research studies [81,82]. 

Among the cytokines analyzed in this study, the levels of IL-8 and IL-6 were significantly increased in the BC in hyperglycemia, compared to both normoglycemic and hypoglycemic environments (groups I and II, respectively). IL-8, also known as neutrophil chemotactic factor or chemokine CXCL8, is a key mediator associated with the immune reaction in the innate immune system response [83]. In relation to IL-6, a cytokine reported to exhibit both inflammatory and anti-inflammatory characteristics, astrocytes are identified as a significant source of this interleukin [84]. While IL-6 can be beneficial in the CNS due to its ability to promote nervous cell growth, an excessive production of IL-6 can be detrimental, contributing to the pathophysiology of CNS disorders [85]. It has been observed that high glucose induces the expression of IL-6 in cultured astrocytes [86].

Additionally, pro-inflammatory cytokines like IL-1α, IL-1β, and IL-2 attained their peak levels in the BC under conditions that mimicked high blood sugar, in contrast to normal blood sugar levels. Previous research, conducted in various systems beyond the BBB, has consistently demonstrated that elevated glucose concentrations lead to an increase in these cytokines [87,88,89,90]. IL-1β is a key immunoregulatory and pro-inflammatory cytokine and its increase is acknowledged as a vital element in the brain’s structured reaction to neuroinflammation and in the recruitment of white blood cells to the CNS [91,92]. The only exception in the first part of the study (the stage 1), was the level of IL-4, which is known as a pleiotropic anti-inflammatory cytokine that functions mainly by suppressing the pro-inflammatory environment. The highest values for IL-4 was observed in conditions imitating hypoglycemia, intermediate values in normoglycemia, and the lowest values in hyperglycemia. Being an important modulator of the immune system, IL-4 is primarily released by mast cells, Th2 cells, eosinophils, and basophils [93]. IL-4 can have either pro-inflammatory or anti-inflammatory impacts on astrocytes, with the outcome depending on the specific treatment and timing regimen [93,94]. It was reported that IL-4 modulates microglia homeostasis and may attenuate slowly progressing neurodegenerative disorders (e.g., amyotrophic lateral sclerosis). Our results may indicate that anti-inflammatory properties of IL-4 are most severely impaired in a hyperglycemic environment.

In 12 h after 0.2 μM LPS addition, a significant increase in the cytokine concentration was observed only for IL-6 and IL-8 (see Figure 5). It may be explained by the relatively low dose of LPS used in the study [95]. Thus, the 12 h period after LPS addition was probably too short to induce a significant rise of pro-inflammatory cytokine levels. Accordingly, in the subsequent measurements, made another 24 h after LPS addition, the increase in concentration could be observed for almost all cytokines (see Figure 6). This indicates that LPS did induce production of pro-inflammatory cytokines, although in the case of IL-1α, IL-1β, IL-4, and partially in case of IL-2 (hyperglycemic conditions), the time interval necessary for this effect was longer. The strong correlation between LPS administration and a rapid increase in the production of IL-6 and IL-8 has been reported in numerous studies performed to date [96,97,98,99]. 

Therefore, it appears that, at least in a controlled laboratory setting, LPS triggers the generation of various pro-inflammatory cytokines at varying time intervals. This aspect should be kept in mind when designing upcoming experiments using this BBB model. Additionally, it is worth noting that endothelial cells in the co-culture also exhibit reactions to LPS, and one element of future research may be to evaluate both the extent and timing of their inflammatory responses [100]. 

The role of the BBB in maintaining homeostasis is crucial, yet it also poses a significant obstacle for the drugs to reach the brain [101]. Developing drugs to address CNS disorders is exceptionally difficult, primarily because of their low bioavailability in the CNS due to the barrier function of the BBB [102].

The crucial part of the study (stages 3 and 4) was to verify if RSV—as a compound with documented anti-inflammatory properties—may cross the BBB and develop its anti-inflammatory action with the same efficacy at different glucose concentrations in MC. As shown in the previous in vivo studies using a different methodology, RSV’s ability to cross the BBB may play a critical role in neuroprotection from the development of diseases linked to the inflammatory response in the CNS [103].

In one study, significant levels of RSV were found in the rodent’s brain after intraperitoneal injection, reaching its peak at 4 h post-injection [104]. In order to clarify whether any correlation between human and rodent bioavailability exists during intraperitoneal administration of RSV, further intensive analysis is needed. Another in vivo study on RSV bioavailability in humans showed that RSV and its major metabolites were detected in plasma and CSF in patients after taking 500 mg of RSV orally once a day [105]. Despite several encouraging discoveries in human clinical trials, there are still numerous conflicting results, which can be partially attributed to variations in the dosing regimens applied [106]. 

In this in vitro study under conditions mimicking normoglycemia, only 0.61% of the RSV was detected in the BC, following its addition to the MC. This percentage was even lower under hypoglycemia- and hyperglycemia-mimicking conditions (0.53% and 0.55%, respectively). As documented by the cytokine profiles assessed in the BC, the potential of RSV to counteract neuroinflammation in the context of normoglycemia is higher than in hyperglycemia- or hypoglycemia-induced BBB disruption.

Due to the inflammatory response and induced cell damage, conditions of hyper- and hypoglycemia are generally associated with increased BBB permeability compared to normoglycemia. We can speculate that because of RSV’s effects on cell adhesion proteins, its neuroprotective activity may partially compensate the effect of dysglycemia. The specific mechanisms of RSV in maintaining or restoring BBB integrity might explain the observed decrease in RSV levels under hyper- and hypoglycemic conditions relative to normoglycemic conditions.

A future challenge would be to assess the uptake and the kinetics of RSV metabolism in the BC. This could help to determine whether RSV is being metabolized by endothelial cells and/or astrocytes (with this phenomenon increased in abnormal glycemic states in the MC) or if the low RSV recovery in the BC observed in this study is primarily due to limited penetration through the BBB.

It was reported that such a low bioavailability of RSV may reduce its anti-inflammatory properties [107]. Extensive metabolism of orally administered RSV in the liver and in the intestine results in an oral bioavailability lower than 1%. While in vitro studies demonstrate a significant beneficial biological effect of RSV within cells, it is reported that its distribution in tissues is exceedingly limited [108]. The insufficient absorption of RSV by human tissues constitutes an important challenge when it comes to applying fundamental scientific findings to clinical applications [106].

For this reason, numerous studies have aimed at enhancing RSV bioavailability, especially in the CNS. It has been found that RSV derivative (i.e., methylated) compounds can be more effective in crossing the BBB, both in vivo and in vitro, and might be a suitable path for the future research [109,110]. The bioavailability of RSV in the CNS could also be improved through encapsulation in nanoparticles. So far, this compound has been encapsulated within a variety of nanosystems, such as liposomes, lipid nanoparticles, and polymeric nanoparticles [111]. Furthermore, it has been demonstrated that the BBB penetrability of RSV is remarkably increased by its intrathecal administration [112]. This route bypasses the BBB to deliver the polyphenol directly to the CNS. It has been demonstrated to be effective in improving RSV bioavailability in the CNS. On the other hand, oral administration is the most common route for administering RSV due to its convenience and patient compliance. There is no doubt that optimization of the formulation composition and administration route is essential for improving RSV bioavailability and CNS penetration, ultimately enhancing its therapeutic potential in neuroinflammatory conditions.

In this study we analyzed six selected cytokines and not a whole group of more than 30 pro-inflammatory cytokines identified to date. Thus, it cannot be excluded that choosing some other pro-inflammatory cytokines could elicit different results [113].

It is crucial to emphasize that the outcomes we obtained are significantly affected by the specific experimental in vitro model we used. First and foremost, in our approach, the circulating blood in the vascular system was replaced with the EBM-2 medium. In this system, the lack of blood results in the elimination of variables influencing the interpretation of the results, including number of cytokines, cells (e.g., activated leukocytes, platelets), and hormones (e.g., insulin, sex steroids). Therefore, it is feasible to provide a simplified explanation of the results, which could be advantageous in highlighting the straightforward connections observed between glucose levels, LPS, RSV, and cytokine production in co-cultured endothelial cells in the MC and astrocytes in the BC. However, this approach leads to an incomplete model and reduces the possibility of directly applying these findings to real-world situations or using them in clinical studies.

Reproducing the results obtained from our in vitro study in animal models would be an essential step in validating the findings and understanding the broader implications. Animal models allow for the assessment of RSV’s effects in a more holistic environment, considering interactions between various cell types, tissues, and organs within a living organism. Additionally, animal models enable us to study the pharmacokinetics, bioavailability, and potential adverse effects of RSV.

Considering sex as a biological factor is also important, as sex differences can influence various physiological processes, including inflammation and glucose metabolism [114]. In animal models, both male and female subjects should be included to assess potential sex-specific differences in response to RSV treatment and under diabetic conditions. Differences in hormone levels, immune responses, and metabolic pathways between males and females may result in varying outcomes [115]. In control and diabetic animal models, it is possible that similar trends in RSV’s anti-inflammatory effects may be observed in both males and females. However, sex-specific differences in response to RSV treatment and diabetic conditions could also be present, potentially affecting the magnitude or direction of the observed effects. Therefore, including both sexes in animal studies would provide a more comprehensive understanding of RSV’s effects and their relevance across different populations.

Overall, while the in vitro study provides valuable preliminary data, further validation in animal models, considering sex as a biological factor, would be essential for advancing our understanding of RSV’s potential therapeutic effects on brain inflammation and glucose metabolism. 

Furthermore, diabetes, a multifaceted metabolic disorder, contrasts with artificially induced high blood sugar situations. Aberrations in the microvascular network of diabetic individuals, characterized by unusual blood vessel growth and heightened vascular density, could potentially affect the levels of cytokines and the activity of RSV.

The effects of multiple doses of glucose and RSV on cytokine production and inhibition, as well as RSV availability in the CNS, were not explicitly explored here. The study focused on investigating the anti-inflammatory effects of RSV on astrocyte cytokine profiles within an in vitro model of the BBB under varying glucose concentrations (2.2, 5.0, and 25.0 mmol/L). To optimize cytokine production inhibition and RSV availability in the CNS, future studies could explore different doses of both glucose and RSV. By systematically varying the concentrations of glucose and RSV, researchers could identify optimal conditions for minimizing cytokine production and maximizing RSV availability. This approach would provide insights into the dose–response relationships of glucose and RSV in modulating inflammation and could develop more effective therapeutic strategies for neurological disorders. 

In conclusion, because the cytokine profile in the CNS is associated with glucose level fluctuations, we investigated if treatment with RSV is beneficial for diabetic patients with neuroinflammation. Supplementary treatment with this polyphenol in diabetic patients with neurodegenerative diseases may improve endothelial function, microvascular blood flow and metabolism, and—most importantly—reduce neuroinflammation. The obtained results suggest that the anti-inflammatory effects of RSV acting across the BBB are expected to be optimal in patients with adequate glucose control, ensuring normoglycemic conditions. For this reason, any therapeutic attempts with RSV in diabetes should be performed after or in parallel to restoring normoglycemia. Future in vivo studies in animals and human tests are needed, including those considering safety issues and side effects, especially excluding RSV toxicity. When establishing the ideal dosage for this substance, it is crucial to account for the biphasic dose–response of RSV in various human cell lines. This biphasic response entails positive effects at lower doses and detrimental effects at higher doses, making it a vital factor to consider. Implementation of the BBB in the in vitro model used in our study might be useful in future studies on RSV bioavailability, considering dose–response studies or using factors enhancing BBB penetration [116]. 

## 4. Materials and Methods

The study was performed between January 2021 and March 2022 in the Warsaw Medical University Center of Preclinical Research and Technology (CePT). 

The study was conducted on commercially available human cell lines not requiring bioethical approval; nevertheless, we obtained approval from the Bioethics Committee from the Medical University of Warsaw on 14 December 2020 (No. AKBE220/2020). 

The BBB model used in this study contained co-cultures of endothelial cells and astrocytes at passages 6–10. A detailed description of all procedures involved in the preparation of this in vitro model is provided elsewhere [117]. Endothelial cells and astrocytes were co-cultured, but they did not make direct contact due to a separation membrane (0.4 µm wide pores). Human cerebral microvascular endothelial cell line hCMEC/D3 was purchased from Cedarlane Cellutions Biosystems (Burlington, ON, Canada; catalogue # CLU512). This cell line is suitable for investigating human neurological pathways, as stated by the manufacturer. Endothelial cells were grown for 3 days to obtain 100% growth confluence in EBM-2 medium (Merck KGaA, Darmstadt, Germany; cat. # C-22211) supplemented with 5% Fetal Bovine Serum, 1% Chemically Defined Lipid Concentrate, 28.5 μM Ascorbic Acid, 62.5 μM bFGF- human Basic Fibroblast Growth Factor, 10 mM HEPES, 1.4 μM Hydrocortisone, and 100.000 U/L Penicillin-Streptomycin (all purchased from Thermo Fisher Scientific-Gibco^TM^; Waltham, MA, USA). The cells proliferated in 5 special culture bottles, each having a surface of 75 square centimeters, provided by TPP, Techno Plastic Products AG (Trasadingen, Switzerland). Normal human astrocytes derived from healthy brain tissue were purchased from Thermo Fisher Scientific-Gibco^TM^ (cat. # N7805100), together with Gibco^TM^ Astrocyte Medium (cat. # A1261301), in which the cells were cultured for 4 days until they differentiated to glial fibrillary acidic protein (GFAP)-positive cells with a typical star-like morphology and reached 100% growth confluence in DMEM Gibco astrocyte medium, containing two additives: 1% N-2 Supplement and 10% One Shot Fetal Bovine Serum (FBS). The astrocytes were cultured in 5 special culture bottles, with 25 square centimeters of surface each, obtained from TPP Techno Plastic Products AG. After obtaining 100% confluence of both cell lines (astrocytes and endothelial cells), the cells were detached from the culture bottle and counted using a trypan blue dye. A mix of 10 µL of a cell suspension and 10 µL of trypan blue dye was placed in chamber’s slide and subsequently in a Countess Automated Cell Counter (Thermo Fisher Scientific-Invitrogen). Each bottle contained about 1 million astrocytes per 1 mL of medium (5 million cells/flask) and about 2 millions of endothelial cells per 1 mL of medium (20 million cells/flask). Afterward, astrocytes were seeded into 24-well plates (Thermo Fisher Scientific-Gibco; catalog number A15690601). The following day, endothelial cells were introduced into insert wells, each with a see-through polyester (PET) membrane with a 0.4 µm pore size and a pore density of 2 × 10^6^ cm^2^. The culture surface area per individual well was 33.6 mm^2^ (Greiner Bio-One GmbH—ThinCertTM, Frickenhausen, Germany; catalog number 662641). Nine 24-well plates were used in this study, with a total number of 216 wells. Each well was covered with Gibco Geltrex Matrix, a basement membrane extract with reduced growth factor properties, specifically employed for the adherence and long-term support of human cells. Astrocyte suspension (3 × 104 cells) was placed in each well and after 24 h the medium was exchanged. Afterwards, the inserts were coated with rat tail-derived type I collagen (Merck C7661; cat. # C7661) and placed in the wells. After the collagen had solidified to a gel-like state, a suspension of 6 × 10^4^ endothelial cells was introduced into every individual insert. Following another 24 h period, the solution was replaced with a new one that had a pre-established glucose concentration. The obtained model of the BBB contained astrocytes representing the CNS side (the brain compartment, BC) and endothelial cells representing the circulatory side (the microvascular compartment, MC), as shown schematically in Figure 8. The cells were cultured in an incubator with controlled humidity at a temperature of 37 °C and a CO_2_ concentration of 5%. 

The integrity of the endothelial cell layer, corresponding to the integrity of the BBB model, was checked indirectly by assessing cell viability. At the stage of developing the BBB model, a correlation was shown between transendothelial electrical resistance (TEER) measurement values indicating loss of integrity and a reduction in cell vitality of more than 25% (compared to the average in a given assay). In the control (reference) group, the average TEER results obtained at 2 h intervals were for periods 1–12, 13–24, and 25–36 h, respectively: 140.4 ± 11, 121.2 ± 09, and 111.6 ± 10 (Ω.cm^2^ ± SEM). The decrease in endothelial cell vitality above 25% correlated with a decrease in the TEER value by at least 30%, compared to the reference group, on average by 30.03 ± 4.7, 31.4 ± 4.5, and 33.6 ± 5.2 (% SEM) for the periods 1–12, 13–24, and 25–36 h, respectively.

The basis for adopting a specific time frame for conducting the experiment, including the intervals for administering specific doses of the substances used, was data, both our own and in the literature, regarding the dynamics of changes in cytokine profiles during the LPS-induced inflammatory reaction and the speed of manifestation of the anti-inflammatory effect of RSV in in vitro culture conditions. On the other hand, the duration of the experiment took into account the possibility of maintaining the stability (integrity) of the BBB model we used.

The three study groups were established, differing in glucose concentrations in the MC. The culture media administered into the MC within the wells of group I, group II, and group III contained D-glucose concentrations mimicking hypo-, normo-, and hyperglycemia (40 [2.2], 90 [5.0], and 450 [25.0] mg/dL [mmol/L], respectively). To verify if glucose concentration itself affects inflammatory response, various cytokine levels in the BC were measured 24 h after glucose addition (**stage 1 of the study**). For each cytokine, six independent samples were analyzed.

The following inflammatory cytokine levels were measured: IL-1 α, IL-1 β, IL- 2, IL-4. IL-6, IL-8, with the Multi-Analyte ELISArray Kit (Qiagen-Q4Lab, Hilden, Germany; cat. # 336161), according to the instructions of the manufacturer. A multi-well ELISA microplate, covered with a variety of antibodies tailored to specific targets, enabled the capture of diverse cytokines. Every ELISArray microplate carried biological samples, in addition to positive and negative control samples. The ELISA test was carried out strictly according to the manufacturer’s protocol. The absorbance was determined at a 450 nm and 570 nm wavelength. Subsequently, the recorded values at 570 nm were subtracted from those at 450 nm when wavelength correction was possible. This measurement was conducted using a monochromator-based microplate reader, the Biochrom Asys UVM 340. This device, manufactured by Biochrom-Harvard Bioscience in Holliston, MA, USA, has a wavelength range of 340–800 nm and can measure optical density (OD) in the range of 0–3.2. The manufacturer specifies the reproducibility of this instrument as 0.8%, with an accuracy of 0.5% from 0.1 to 1.0 OD at 450 nm and 1.0% from 1.0 to 2.0 OD at 450 nm, with deviations of 0.005 OD and 0.010 OD, respectively. Six repetitions of each measurement were conducted, and subsequently, the mean values in absorbance units (Au) were determined for the specified cytokine concentration, with the results expressed as pg/mL ± SD (pg/mL ± SD). The negative control samples, which were in the form of Sample Dilution Buffer, were placed in the initial row of the microplate, denoted as row A. The positive control samples, consisting of the Antigen Standard Cocktail, were positioned in the final row, specifically designated as row H. 

In this study, a solution of LPS with a final concentration of 0.2 μM was introduced to the culture plates forming the BBB on the side of the BC to create conditions resembling neuroinflammation. After 12 and 36 h, the ELISA was used to examine the cytokine profiles, including IL-1 α, IL-1 β, IL-2, IL-4, IL-6, and IL-8, in various study groups (labeled as groups I–III). These groups had different levels of glucose in the culture medium. Each measurement of cytokine levels represented the average value derived from six determinations in the experimental samples obtained from the brain cells. These activities constituted the **2nd phase of the study**.

A study to investigate how LPS treatment affects the shape of astrocytic cells cultured under different glucose conditions was conducted. After a 36 h culture period in a model simulating the BBB, three study groups were established, with some receiving LPS and others not. These cells were fixed in a 3% paraformaldehyde (PFA) solution in PBS for 30 min at room temperature. Subsequently, the cells were embedded in paraffin and stained with hematoxylin and eosin (H&E).

To examine the cells, a Zeiss Primovert inverted cell culture microscope was used, equipped with various light sources, including HAL 35 W and a 3W LED with infinity optics, along with a Zeiss Axiocam 105 Color camera for image capture. The acquired images using ZEN 2.3 software were analyzed.

Furthermore, the paraffin-embedded sections under a light microscope (Leica DM 400B) with the expertise of two experienced neuropathologists were examined. Each group was assessed using 50 images. High-resolution photographs of the H&E-stained astrocytes were taken with a camera attached to the microscope. The purpose of this preliminary assessment was to determine whether there was a valid reason for employing computer image analysis procedures.

To investigate how RSV affects the inflammatory response 36 h after introducing LPS, RSV solution with a final concentration of 50 μM was applied to each sample in the MC. Following this, an immunoenzymatic assay, specifically the Multi-Analyte ELISArray Kit, was employed to quantify the levels of cytokines such as IL-1 α, IL-1 β, IL-2, IL-4, IL-6, and IL-8 after a 24 h interval, signifying the conclusion of **the 3rd stage of the study**. Each measurement was performed six times, and subsequently, the average values were subjected to statistical analysis.

Finally (**stage 4 of the study**), transfer of RSV between the compartments was measured by the comparison of RSV concentration in the BC between the studied hypo-, normo-, and hyperglycemic groups (groups I-III) 12 and 24 h after administration of LPS. The ELISA kit (MyBioSource, Inc., San Diego, CA, USA; cat. # MBS2700660) was used. All of the measurements were performed six times, and the mean values were statistically analyzed. 

To ensure the research model (BBB) was validated and the results could be correctly understood, suitable control groups were designed for each phase of the study, as outlined in Table 1.

In the initial phase of the study, a control group was established using a basal medium devoid of any glucose concentration.

During the subsequent stage of the study, which involved the addition of LPS, another control group was formed. This control group utilized a basal medium with 0.2 μM LPS but excluded any glucose solution.

In the final phase of the study, a control group was designed with a basal medium containing 0.2 μM LPS and 50 μM RSV but devoid of glucose. Each control group underwent measurements six times to ensure robust data collection and analysis.

After obtaining the results and verifying the completeness of the data, the normal distribution of the results in all study groups was verified. We used the R programming language and its inherent capabilities (The R Project for Statistical Computing; version 4.0.5) for statistical analysis, in addition to employing the Mann–Whitney U test. The results are presented as the average values with their corresponding standard deviations (SD), and statistical significance was determined if the *p*-value was lower than 0.05.

## 5. Conclusions

In our study, RSV exhibited anti-inflammatory action by decreasing the concentration of all of the pro-inflammatory cytokines (with significant decreases noted for most of them) in all three study groups, corresponding to different levels of systemic glycemia. Regardless of the co-existing glucose concentration in the MC, the cytokine concentrations in the BC were lower in RSV-administered samples compared to RSV-free samples. Importantly, the largest decrease was observed in normoglycemia-simulating conditions (group II) for IL-1β, IL-6, and IL-8 (*p* < 0.05). It suggests that normoglycemia provides an optimal environment for anti-inflammatory action of polyphenols such as RSV. The cytokine concentrations in the normoglycemic environment were decreased to various extents, with the most pronounced effect on IL-1β, but also with great impact on IL-8. The hitherto obtained results suggest a correlation between RSV efficacy and normal glucose concentration in MC for most analyzed cytokines, although the concentration of the only anti-inflammatory cytokine, IL-4, was the most affected by RSV in hypoglycemia conditions. 

Thus, referring to the anti-inflammatory effects of RSV at the level of the CNS, the importance of adequate glycemic control and restoring normoglycemia in diabetic patients with diseases and conditions such as Alzheimer’s disease, Parkinson’s disease, stroke, multiple sclerosis, and traumatic brain injury should be appreciated. Anti-inflammatory agents such as RSV could most effectively perform their functions in the normal range of glucose concentration, which results, among others, in the optimal neuroprotection and reduction in neuroinflammation. Further in vivo studies on this flavonoid are worth conducting to evaluate its neuroinflammation-preventing effect, especially in patients with diabetes.

## Figures and Tables

**Figure 1 ijms-25-03110-f001:**
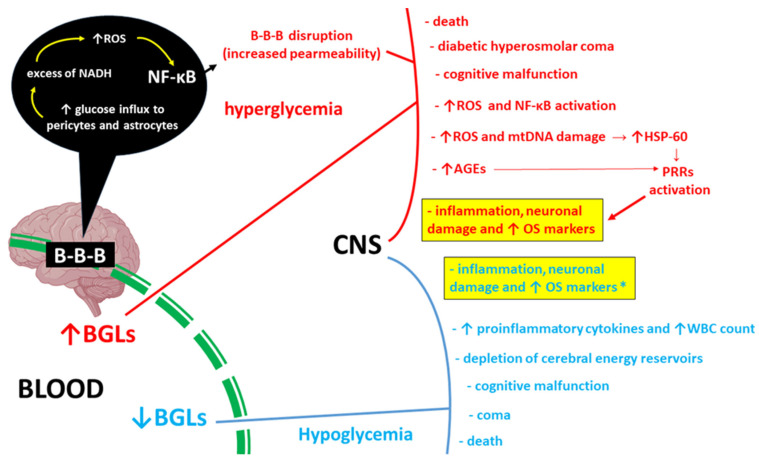
Pro-inflammatory consequences of hyper- and hypoglycemia (marked in red and blue, respectively) in the central nervous system (CNS). Increased glucose inflow into pericytes and astrocytes in hyperglycemia leads to excess reduced nicotinamide adenine dinucleotide (NADH), which results in excessive reactive oxygen species (ROS) production and activation of nuclear factor kappa-light-chain-enhancer of activated B cells (NF-κB) [3,11,17]. This damages the integrity of the blood–brain barrier (B-B-B) and increases its permeability [18,19,20]. Disturbance of glucose homeostasis in the cerebral compartment leads to the induction of inflammation in a similar mechanism. Additionally, mitochondrial DNA (mtDNA) damage is accompanied by an increase in the production of the chaperone protein HSP-60, which can activate pattern recognition receptors (PRRs) and subsequently induce innate immunity-related inflammation [23]. Advanced glycation end products (AGEs) also activate PPRs [31]. CNS inflammation in iatrogenic hypoglycemic states in type 2 diabetes (T2D) is accompanied by increased concentrations of proinflammatory cytokines and elevated white blood cell (WBC) count. Both hyper- and hypoglycemia, inflammation, and increased concentrations of oxidative stress (OS) markers are accompanied by clinical symptoms of varying severity [32]. ** in relation to iatrogenic hypoglycemia in T2D; ↑ BGLs—increased blood glucose levels; ↓ BGLs—decreased blood glucose levels*.

**Figure 2 ijms-25-03110-f002:**
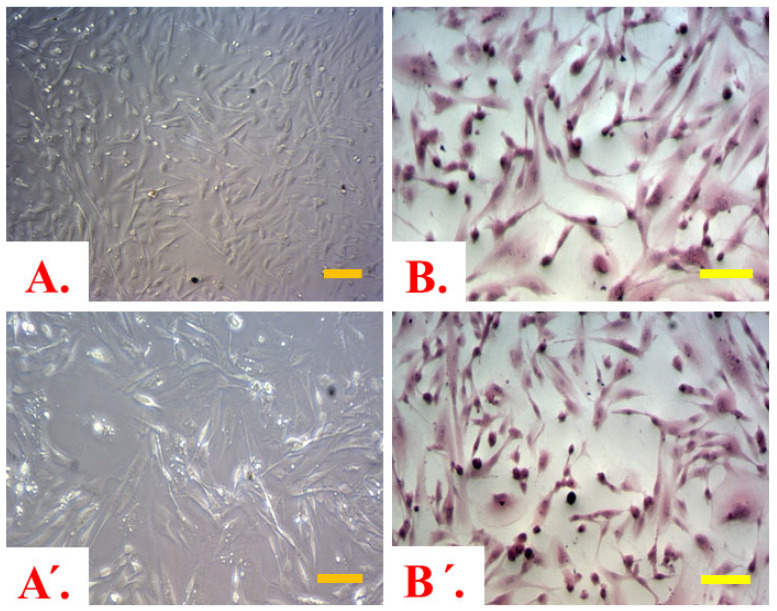
Astrocytes from the brain compartment (BC): (**A**,**A’**)—after glucose serum added and after 0.2 μM LPS solution added, respectively—no staining; (**B**,**B’**)—astrocytes after glucose serum added and after 0.2 μM LPS solution added, respectively—staining with hematoxylin and eosin (H&E). Despite different glucose concentrations used, no significant changes in astrocyte’s morphology have been observed (based on analysis of slides stained with H&E). Scale bars = 50 μm.

**Figure 3 ijms-25-03110-f003:**
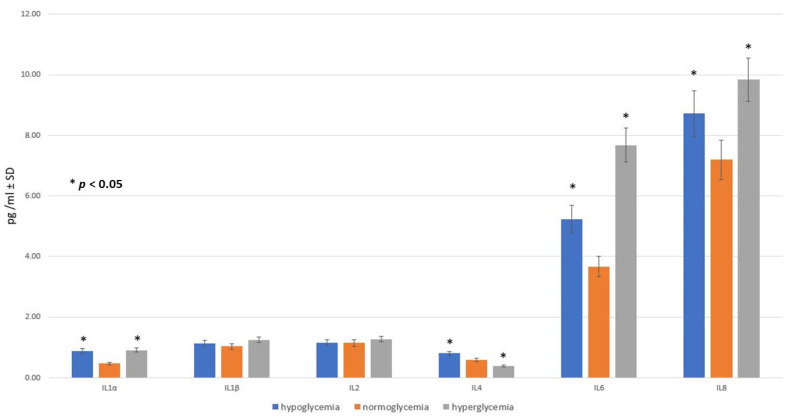
The cytokine concentrations in the brain compartment after 24 h following glucose addition to obtain hypo-, normo-, and hyperglycemic environments: mean values ± standard deviation (pg/mL ± SD). Mann–Whitney U test (*n* = 6 records for each cytokine within each group).

**Figure 4 ijms-25-03110-f004:**
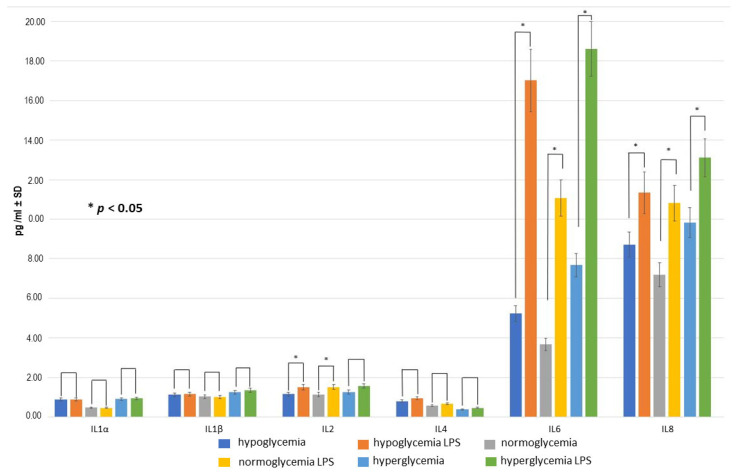
The cytokine concentrations in the brain compartment (BC) after 12 h following 0.2 μM LPS solution administration in BC in hypo-, normo-, and hyperglycemic environments (with 24 h duration of exposure to different glucose concentrations before LPS administration) compared to the respective LPS-free environments at the beginning of experiment: mean values ± standard deviation (pg/mL ± SD). Mann–Whitney U test (*n* = 6 records for each cytokine within each group).

**Figure 5 ijms-25-03110-f005:**
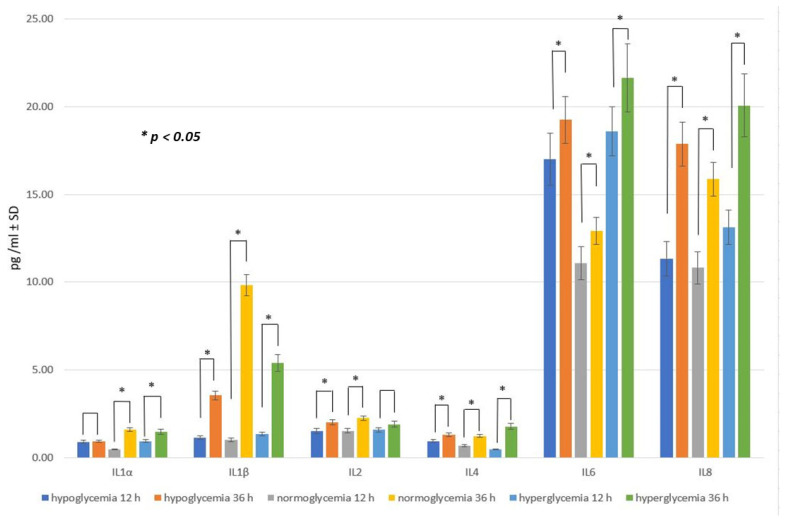
The cytokine concentrations in the brain compartment after 36 h following 0.2 μM LPS solution administration in BC in hypo-, normo- and hyperglycemic environments (with 24 h duration of exposure to different glucose concentrations before LPS administration) compared to the respective values obtained 12 h after 0.2 μM LPS administration: mean values ± standard deviation (pg/mL ± SD). Mann–Whitney U test (*n* = 6 records for each cytokine within each group).

**Figure 6 ijms-25-03110-f006:**
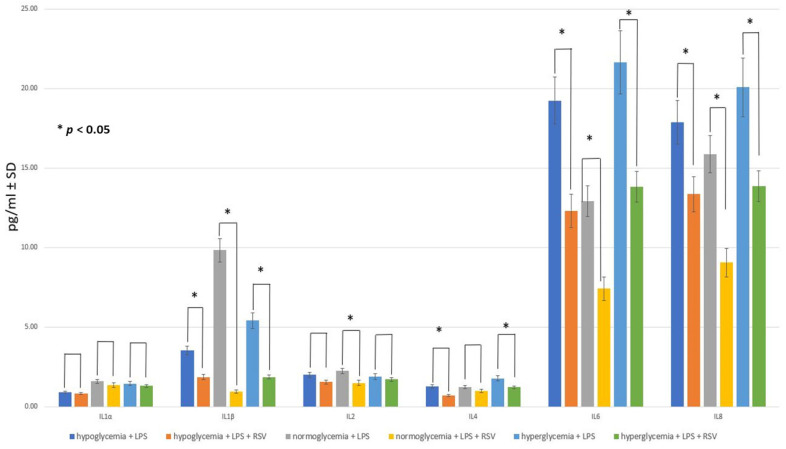
The cytokine concentrations in the brain compartment after 24 h following 50 μM resveratrol (RSV) addition (and after 36 h following 0.2 μM LPS solution adding) into hypo-, normo-, and hyperglycemic environments compared to the respective values obtained after 36 h following 0.2 μM LPS solution addition without subsequent administration of RSV: mean medium values ± standard deviation (pg/mL ± SD). Mann–Whitney U test (*n* = 6 records for each cytokine within each group).

**Figure 7 ijms-25-03110-f007:**
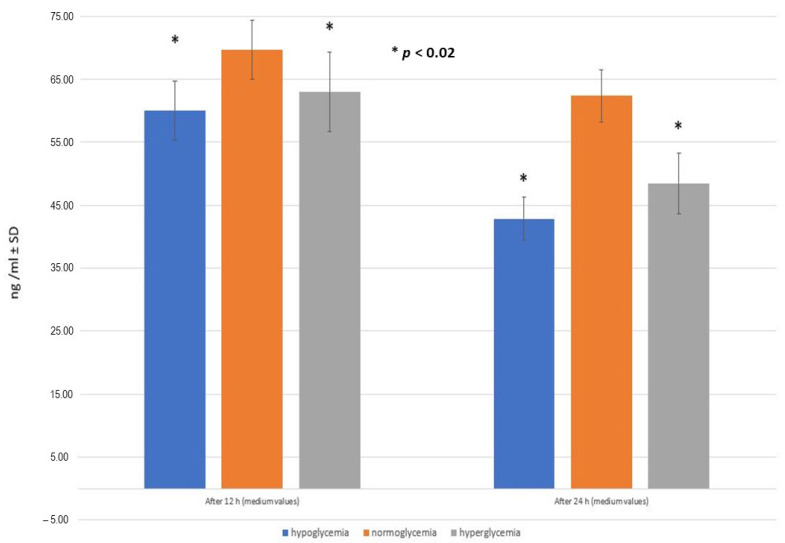
The mean concentration of resveratrol (RSV) in the brain compartment after 12 h and after 24 h following RSV adding into hypo-, normo-, and hyperglycemic environments of the microvascular compartment (ng/mL ± SD). Mann–Whitney U test (*n* = 6 records for each timepoint within the groups).

**Figure 8 ijms-25-03110-f008:**
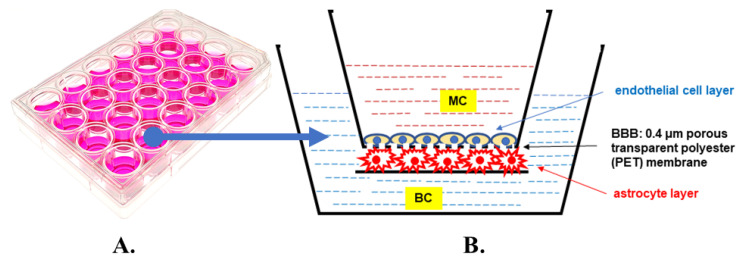
Multi-well plate used for co-culture of endothelial cells and astrocytes (**A**) and the schematic cross-section of the single well illustrating the blood–brain barrier (BBB) model used in this study (**B**). For a detailed description, see Section 4.

**Table 1 ijms-25-03110-t001:** The set of controls: content of culture media used at various stages of the study in control groups from the microvascular compartment (MC) side of the in vitro model of the blood–brain barrier.

Control Groups	Basal Medium	Glucose Medium	LPS Solution	RSV Solution
1st stage of the study	+	−	−	−
2nd stage of the study	+	−	+	−
3rd stage of the study	+	−	+	+

## Data Availability

The authors declare that the data supporting the findings of this study are available within the paper. Should any raw data files be needed in another format, they are available from the corresponding author upon reasonable request.

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
