# Peer review of "Anti-Inflammatory Action of Resveratrol in the Central Nervous System in Relation to Glucose Concentration—An In Vitro Study on a Blood–Brain Barrier Model"

_ijms, 2024, doi:10.3390/ijms25063110_

Round 1

Reviewer 1 Report

Comments and Suggestions for Authors

Reviewer’s comments

In this review article, "Anti-inflammatory action of resveratrol in the central nervous system in relation to glucose concentration: an in vitro study on a blood-brain barrier model," by Komorowska J et al., the authors have demonstrated the anti-inflammatory role of resveratrol. In addition, the anti-inflammatory effects of RSV are dependent on normoglycemic (2.2 mmol/L)) conditions but not on hypo- and hyperglycemic conditions. They have also demonstrated the time-dependent and LPS-induced cytokine production and inhibition by RSV in control and BBB models.

The authors have nicely discussed and concluded that abnormal blood glucose levels can significantly impair RSV availability in the CNS, promoting neuroinflammation. I have a few questions.

  1. All these studies were performed in a cell culture system. Can these results be reproduced in animal models? If we consider sex as a biological factor, can we expect similar results in both males and females in the control and diabetic animal models?
  2. Did the authors try to use a few lower and higher doses of glucose and RSV to get optimal levels of cytokine production and inhibition and RSV availability in the CNS?

Reviewer 2 Report

Comments and Suggestions for Authors

The current manuscript is an interesting experimental study on the action of resveratrol on neuroinflammation using an in vitro BBB model. It appears to be overall well-done, with many relevant assays having been performed. Hence, I only advise on the following alterations before acceptance for publication:

- The full stop mark currently present at the end of the manuscript’s title should be removed;

- In the abstract, the authors should add an initial sentence explaining why and how glucose levels, resveratrol and brain inflammation are related;

- In the introduction section, some sentences at the end of each paragraph lack the appropriate reference, this should be corrected;

- The connection between glucose levels and central nervous system action mechanism should be schematized in a representative image, which should be added to the introduction section, for better reader visualization;

- The number of experiments (“n” value) and the statistically relevant information should be added on each graph´s caption;

- In the results section, subsection titles should be changed from “Stage” to other more specific designations, namely mentioning what was done and/or what changed in each “Stage”;

- Given the promising results, authors should mention which administration route and which formulation type (and formulation composition) they would use for the administration of resveratrol for the treatment of neuroinflammation;

- Figure 7 quality (resolution) should be improved;

-  The conclusion should include which diseases would be targeted;

- The conclusion section should not include any references;

- An abbreviation list is missing and should be added.
